# Cytogenetic, Morphometric, and Ecological Characterization of *Festuca indigesta* Boiss. in the Southeast of Spain

**DOI:** 10.3390/plants11050693

**Published:** 2022-03-04

**Authors:** Gloria Martínez-Sagarra, Federico Casimiro-Soriguer, Sílvia Castro, João Loureiro, Juan A. Devesa

**Affiliations:** 1Departamento de Botánica, Ecología y Fisiología Vegetal, Campus de Rabanales, Universidad de Córdoba, 14071 Córdoba, Spain; bv1dealj@uco.es; 2Departamento de Botánica y Fisiología Vegetal, Campus de Teatinos, Universidad de Malaga, 29010 Málaga, Spain; fedeque@hotmail.com; 3Centre for Functional Ecology, Department of Life Sciences, University of Coimbra, Calçada Martim de Freitas, 3000-456 Coimbra, Portugal; scastro@bot.uc.pt (S.C.); jloureiro@bot.uc.pt (J.L.)

**Keywords:** *Festuca*, genome size, National Park, Poaceae, Sierra Nevada, Sierra de las Nieves, Spain

## Abstract

*Festuca indigesta* subsp. *indigesta* (Poaceae) is endemic to the southeast of Spain, and until recently, it was considered that its range of distribution was restricted to the siliceous core of Sierra Nevada. However, it has been recently extended in the territory to others calcareous mountains. This study investigates the cytogenetic variability throughout the geographic range of this taxon, the possible edaphic preferences of each cytotype, and the morphological variation of cytotypes. Genome sizes and ploidy levels were estimated using flow cytometry and chromosome count. Soil samples were collected to test the nature of the substrate, i.e., pH, and calcium and magnesium contents. Finally, morphological characters were measured in herbarium specimens. This study provides the first genome size data for the species. Hidden cytogenetic diversity was detected in the taxon, comprising hexaploid (2n = 6*x* = 42), octoploid (2n = 8*x* = 56) and dodecaploid (2n = 12*x* = 84) individuals. No relationship between substrate nature and cytotype was observed. Morphological differences were detected for the size of floral parts and stomata among cytotypes, but these were blurred if the entire morphological variation range was considered. Our results suggest that each mountain range could act as a reservoir of morphologically cryptic genetic diversity regarding this taxon.

## 1. Introduction

*Festuca* L. (Poaceae) is, from the phylogenetic, morphological, and cytogenetic perspective, considered to be a complex genus, and this has resulted in a difficult taxonomy. It has been suggested that the Western Mediterranean region was one of the secondary centers of diversification of *Festuca* during Pleistocene glacial periods, and one of the main driving forces behind its diversification has been postulated as being polyploidization events [1]. The genus includes many polyploid species (about 70%) and comprises a large variation in cytotypes, ranging from 2n = 2*x* = 14 to 2n = 14*x* = 98; the latter being recently detected in plants from the Iberian Peninsula [2]. In the Iberian territory, a great taxonomic diversity of the genus is concentrated in the Baetic Range (southeast of Spain), which harbors several endemic fescues, some with a narrow distribution area and others that are also present in North Africa (Baetic-North African species). The orographic diversity of the Baetic Cordillera allowed the *Festuca* species and the taxa of many other genera to find suitable niches. This, together with the variability of other abiotic characters, such as extreme temperatures and a wide variety of edaphic conditions, provided numerous opportunities for the occurrence of hybridization and stimulated the emergence of polyploids, leading to the formation of new entities [3]. Climatic history and ecology are considered the most important factors in shaping the spatial pattern of genetic diversity within species [4]. Some studies have shown that substrate type affinity represents a major driver of plant evolutionary diversification [4,5,6]. This fact is quite evident in the genus *Festuca*, as many endemic species show a strong preference for either calcareous or siliceous substrate (substrate specificity) [7]. In the Baetic mountain range, most of these taxa tend to be concentrated in disjunct mountains, also called “highland islands” [8], where the vegetation is composed mainly of xerophytic shrubs and open hard-leaved grasslands on poorly developed calcareous or siliceous soils. Not surprisingly, this territory is one of the most important hotspots of plant diversity in the Mediterranean Basin, with the two main endemic centers being Sierra Nevada and Serranía de Ronda [9]. 

*Festuca indigesta* Boiss. (*Festuca* sect. *Festuca*) is a polyploid Iberian−Maghreb species, which, in the Iberian Peninsula, is represented by the subsp. *indigesta*. This taxon has great ecological importance and dominates many plant communities that characterize the high mountains of southeastern Spain, where it is endemic [7]. Since it was described by Boissier [10], it has been assumed that the taxon is distributed throughout the siliceous oro-mediterranean bioclimatic belt (at 1800–2800 m.a.s.l.) of the Sierra Nevada National Park (provinces of Granada and Almería). In terms of lithology, this area mostly consists of dark schist and feldspar-bearing mica-schists from the Nevado-Filabride metamorphic complex [11]. However, this taxon was recently also found in calcareous mountains (Figure 1), specifically the high mountains of the Sierra de Gádor (the easternmost population) [12], and in the disjunct Sierra de las Nieves National Park (Serranía de Ronda, province of Málaga), which constitutes the limit of its westernmost distribution [7]. The latter population is the most isolated, and is located 140 kilometers to the west of the most eastern population (Figure 1). The subspecies has also been detected in basic outcrops in the Nevada core, i.e., in the Collado de Las Sabinas population. 

Closely related polyploid fescues (*Festuca* sect. *Festuca*) that grow in other mountain systems of the Iberian Peninsula (Pyrenees, Cantabrian range, Iberian System, and Sistema Central mountain range) have been grouped in the “*indigesta* complex” by several authors, owing to their morphological similarity with *F. indigesta* subsp. *indigesta* (i.e., plants densely tufted with intravaginal innovation shoots, leaf sheets with overlapping not closed margins, and sclerenchyma arranged in a thick continuous ring in the leaf cross-section, among other traits) and to the assumption that they all grow exclusively on siliceous substrates [13,14,15]. However, according to the current taxonomic treatment published in Flora iberica [7], most of these plants belong to *F. yvesii* Sennen and Pau, from which *Festuca indigesta* subsp. *indigesta* can be distinguished easily by having a pungent leaf apex (i.e., terminating gradually in a hard, sharp point), while in the case of *Festuca yvesii*, it may be more or less acute, but never pungent. Furthermore, regarding the apparent substrate specificity of these species, it has recently been suggested that both taxa could be more generalist than expected, being able to grow on different types of substrates [7]. It does not appear to be very difficult to correctly identify *F. indigesta* subsp. *indigesta*, owing to this qualitative character, and this is possibly why less attention has been paid to its morphological variation in quantitative terms when compared with other groups in which small variations in the morphological range are essential for a correct identification. However, our preliminary morphological analyses at the population level have revealed some variability among populations regarding the size of the floral parts.

Genome size and ploidy level can be additional traits in the characterization of a taxa, and are often related to differences in the morphology, ecology, evolution, or distribution [16], as has been detected in some fescue taxa [17,18,19]. The chromosome counts carried out to date have shown a homogeneous hexaploid level (2n = 6*x* = 42) for the species, with a single octoploid (2n = 8*x* = 56) count in a supposedly mixed-ploidy population [20]. These counts were carried out on plants grown on acidic substrates in two populations in Sierra Nevada. However, there is no karyological information for the plants that grow on calcareous substrates.

Considering all this, we assessed genome size and ploidy level variation, ecological/edaphic preferences, and morphological variation in *F. indigesta* subsp. *indigesta* throughout its distribution range. Our main goals were (1) to study the cytotype distribution and diversity within and among populations using flow cytometry (FCM) and chromosome counts, (2) to characterize the soil variables in order to verify edaphic preferences and to clarify whether there was any correlation between cytotype and soil type, and (3) to analyze macro- and micro-morphological characters in order to detect a possible relationship with cytotype.

## 2. Results

### 2.1. Genome Size, DNA Ploidy Level, and Chromosome Numbers

Flow cytometry histograms showed a good quality (Figure 2), with a mean sample CV of 4.38% for the material of *Festuca indigesta* subsp. *indigesta* and 3.61% for that of the standards (Table 1). 

Genome size estimates revealed considerable variation among the populations analysed, with an almost 2-fold maximum variation between the smallest (2C = 12.99 pg DNA) and largest (2C = 25.33 pg) mean values (Table 1). Significant differences in genome size estimates were observed among the populations (F_4,26_ = 350.6, *p* < 0.001). Plants from Peñones de San Francisco (2C = 12.56–13.61 pg) and Collado de Las Sabinas (2C = 12.35–14.34 pg) had the smallest genome sizes, followed by those from Puerto de La Ragua (2C = 17.35–18.33 pg) and Sierra de Gádor (2C = 16.59–17.54 pg), while those from the Sierra de las Nieves population had the largest genome sizes (2C = 23.73–26.37 pg; Table 1). 

It was also confirmed by chromosome counts that these genome size estimations corresponded to three ploidy levels (Figure 3): hexaploid (2n = 6*x* = 42 chromosomes; Peñones de San Francisco and Collado de Las Sabinas populations), octoploid (2n = 8*x* = 56; Puerto de la Ragua and Sierra de Gádor populations), and dodecaploid (2n = 12*x* = 84; Sierra de las Nieves population). The 2C values were positively correlated with the chromosome numbers detected (Pearson r = 0.989, *p* < 0.0001; Spearman r = 0.920, *p* < 0.0001). All populations were single-ploidy: no numerical variation in chromosome numbers was found, and there was no significant variation in genome size estimates. 

### 2.2. Soil Analyses and Ecological Remarks

A wide edaphic range was detected in terms of Ca and Mg contents and soil pH (Table 2). The soils of the populations could be grouped into two main groups. Soils from Peñones de San Francisco and Puerto de La Ragua populations were acidic (pH 6.1–6.3) with very low values of calcium (3.4–3.7 meq/100 g) and without free calcium carbonate reacting with the hydrochloric acid, as can be expected with siliceous rocks such as mica-schists. The soils of the remaining populations (Sierra de Gádor and Sierra de las Nieves; Table 2) were alkaline (pH 7.4–8.3), with a superior Ca content (15.1–42.62 meq/100 g) when compared to acidic soils (3.4–3.7 meq/100 g), and a high calcium carbonate content (from 10% to 25%), as is typical of calcareous rocks such as limestone or dolomite. In the Collado de Las Sabinas population, the soils were also alkaline but had a moderate Ca content (7.9 meq/100 g), and no effervescence was observed after treatment with 1 M HCl, despite being marble rocks. As expected, the dolomite populations had high Ca and Mg contents in the soils (e.g., Sierra de Gádor population). Mg values were also high in two soil samples from Sierra de las Nieves population (4.03–4.82 meq/100 g).

With respect to the ecology of the subspecies, it grows in open habitats, forest fringes and clearings, rocky shelves, and stony soils, where it is part of grasslands and xerophytic scrubs. The subspecies participates in the organization of vegetation ecosystem of the oro--Mediterranean belt. In the acidic central core of the Sierra Nevada Mountain Range, it grows in psychro-xerophilous siliceous grassland at elevations of 1800–2800 m.a.s.l., reaching the limit with the crioro-Mediterranean belt. In the case of Collado de Las Sabinas (at 2195–2170 m.a.s.l.), Sierra de Gádor (at 1900–2247 m.a.s.l.), and Sierra de las Nieves (at 1350–1800 m.a.s.l.), the typical calcareous and dolomitophilous xeroacanthic vegetation was found growing with *F. indigesta* subsp. *indigesta*. The subspecies can also grow in Sierra de las Nieves, exceptionally, at lower elevations (1350–1400 m.a.s.l.), in the supra-Mediterranean bioclimatic belt.

### 2.3. Morphological Analyses

As a whole, the sizes of the characters increased according to the ploidy level (6*x* < 8*x* < 12*x*), although there was considerable overlap in the ranges of morphological values (Figure 4). Statistical differences were found among cytotypes for most of the characters studied (Figure 4), with the exception of the vegetative characters, i.e., culm, leaf diameter, auricle, and awn lengths (Appendix A). The highest ranges of size variation for floral characters and stomata were found at the 12*x* level, and the lowest at the 6*x* level (Figure 4). The spikelet, glumes, lemma, and anther lengths, and the lengths of the accessory cells and guard cells of the stomata were significantly different (*p* < 0.05) among the cytotype pairs studied (Figure 4 and Appendix A). The length of the upper glume was significantly correlated with the length of the lemma (Spearman’s rho = 0.81, *p* < 0.001). The same was observed for the lower glume with the upper glume (Spearman’s rho = 0.74, *p* < 0.001), and for the length of the accessory cells of the stomata when compared with that of the guard cells (Spearman’s rho = 0.85, *p* < 0.001). 

In the PCA diagram, both the first and second components contributed to the differentiation of cytotypes, with the first and second component explaining 61.1% and 13.4% of the variation, respectively (Figure 5). The highest eigenvalues obtained in the analysis were concentrated in the first component, and the characters with the highest correlation were those related to the sizes of the floral parts: lemma, upper glume, and spikelet lengths; those mostly correlated with the second axis were the accessory and guard cells of the stomata (Figure 5A and Appendix A). In the first component of the PCA, octoploids strongly overlapped, in an intermediate position, with hexaploids and dodecaploids (Figure 5B). A somewhat greater morphological differentiation was noted between hexaploids and dodecaploids; however, both cytotypes showed considerable overlap of confidence ellipses.

## 3. Discussion

### 3.1. Genome Size, DNA Ploidy Level, and Chromosome Numbers

This study shows that *F. indigesta* subsp. *indigesta* is formed of a polyploid series that comprises hexaploid, octoploid, and dodecaploid individuals. Moreover, genome size estimations confirmed these cytotypes, with values (ranging from 12.99 to 25.33 pg/2C), which were quite similar to those obtained in closer *Festuca* taxa [2,21,22,23]. Our study provides additional data to those previously carried out and supports flow cytometry as a reliable method to estimate ploidy levels in the *Festuca* genus. It is not uncommon to find cytogenetic variability at the intraspecific level in the Poaceae family (e.g., [18,20,21,22,23]), and different chromosome numbers have even been found in subspecies of *Festuca* with great taxonomic complexity [2,20,21]. In addition to the already known hexaploid level [24,25,26], the octoploid level was detected in our study. The only previous octoploid count was reported in a single plant from the Puerto de La Ragua population, in which hexaploid plants dominated [20]. However, the octoploid plants were found to a greater extent than previously assumed, as we found the octoploid level in all the samples from Puerto de La Ragua, and in those from the Sierra de Gádor, for which no cytogenetic data have been available until now. Interestingly, in both populations the taxon coexists with *F. longiauriculata* Fuente, Ortúñez & Ferrero Lom. (*Festuca* sect. *Festuca*), a related diploid taxon (2n = 2*x* = 14) that is more delicate than *Festuca indigesta* subsp. *indigesta*, from which it differs by having smaller spikelets, glumes, lemmas, and anthers [7]. Fuente et al. [20] speculated that the octoploid plant they found could have an allopolyploid origin, and its prevalence in the populations detected may indicate that it has stabilized. However, studies addressing the origin of these populations of *F. indigesta* subsp. *indigesta*, and whether they represent allo- or auto-polyploids, are still lacking. Whatever the case may be, these polyploids have a niche divergence with respect to the diploids of *F. longiauriculata*, which are mostly concentrated in the northeastern parts of Sierra de los Filabres and Sierra de Baza (Figure 1), where *F. indigesta* subsp. *indigesta* has not been found to date [7]. The dodecaploid level is new to the species and was found in the western populations. These data represent the second record of this ploidy level in the genus (the first was detected in *F. yvesii* subsp. *summilusitana* [2]). The decaploid level that would complete the chromosome series (2n = 6*x*–12*x*) has not been found in our study.

### 3.2. Cytotype Distribution and Ecological Parameters

A geographical distribution pattern of the cytotypes was observed. The easternmost part of the Baetic mountain range hosts a mosaic of hexaploid and octoploid populations, with the hexaploids occupying a narrower niche in the Nevadense core, growing in both acidic and basic substrates, with the octoploids having a more extensive distribution to the east, and also growing on both types of substrates, while the dodecaploids exclusively inhabit the western disjunct mountains, growing on limestone and dolomite. All populations showed cytotype uniformity (i.e., no mixed ploidy populations were detected). Hexaploids and octoploids are restricted to higher elevations than dodecaploids, which had broader adaptability at lower elevations. Although the chemical composition of the rocks can trigger evolutionary divergence and have an important influence on the flora and vegetation [4,5,6,27,28], the distribution and diversity of cytotypes of *F. indigesta* subsp. *indigesta* did not show any patterns associated with the substrate nature (or the lithological group). Hexaploids and octoploids were able to grow on siliceous and calcareous bedrock, although dodecaploids were only found on calcareous bedrock. The lack of cytotype-specific associations with substrate type has also been detected in other plant groups (e.g., *Brassica arenosa* group) in which random colonization processes and genetic drift could have played a role in cytotype segregation [29]. In fact, this parameter could be less restrictive for the presence and growth of *F. indigesta* subsp. *indigesta* than has previously been assumed, as it is capable of growing in a wide range of edaphic conditions. The limiting factors for the survival of the subspecies are probably the bioclimatic profile (i.e., elevational gradient) and the habitat type, as the subspecies occurs at high-elevations, and in xeric and wind-exposed ecosystems in which it is highly competitive. These types of ecosystems are particularly sensitive to slight climatic changes and to different types of human impact or pressures [30], signifying that it is necessary to pay attention to and monitor the populations of this taxon. Given the peculiarity and fragility of these ecosystems, germplasm banks and conservation plans should consider the entire cytogenetic variability detected for the subspecies.

### 3.3. Morphological Variation among Cytotypes

Polyploidy is positively related to the size of the morphological characters in many plant groups [31]. Many of the phenotypic changes can be explained by increased cell size relative to cell number, leading to increased organ size. However, the morphological differences among cytotypes of the same taxon (species or subspecies) are, in many cases, not very clear, as has been seen previously in the genus *Festuca* [2,18] and in many other complex taxonomic groups [32,33]. In this study, the morphological differences detected among the ploidy levels (6*x*, 8*x*, and 12*x*) are not sufficiently great to be able to fully distinguish the cytotypes, as a broad overlap among cytotypes was observed, even for the most discriminant morphological characters (e.g., lengths of spikelet, lemma, glumes, anthers, and stomata). However, a notable morphological difference between hexaploid and dodecaploid individuals was observed, with octoploids being difficult to distinguish from the former cytotypes owing to an overlapping gradient of variation. As other morphometric studies indicated, there are high levels of phenotypic variation in the species of *Festuca*, which are probably related to the relatively recent diversification suggested by molecular studies (in the Pleistocene, ca. 2 Ma [34]) and with an ongoing differentiation [2,7]. Although the cytotypes vary slightly regarding the size of the characters, all of them are easily assignable to the same species owing to one qualitative character: the pungency of their leaves, which is particularly noticeable in the plants from the eastern populations (Sierra Nevada and Sierra de Gádor). The length of the stomatal guard cell pairs has been used to predict ploidy level in many groups of grasses [35,36], but the wide variation in the size within each ploidy level makes the use of this inference very difficult in the case of *Festuca*. Moreover, the size (and density) of the stomata may also be associated with environmental variables, which may mask the effect of whole genome duplications [37,38]. The plants from the Sierra de las Nieves population (2n = 12*x* = 84) were the most discordant morphologically, with broader ranges regarding certain characters, especially the culm size (Appendix A). This could be associated with not only a greater potential for phenotype variation in the highest polyploids, but also the fact that the plants from this population have less pungent leaves and are, therefore, more palatable and subjected to greater grazing intensity than what occurs in the eastern populations. Despite this, these plants cannot be easily differentiated by merely using morphological characters. In future scenarios, the allopatric phenomena produced by the biogeographic isolation of this atypical population could lead to a new taxonomic entity. The cytotype diversity found throughout the distribution range of *F. indigesta* subsp. *indigesta*, coupled with the absence of spatial overlap among cytotypes at the intrapopulation level, suggests that the mountains of the Baetic Cordillera may act as evolutionary laboratories that could play a significant role in promoting the genetic diversification of the taxon, which is not clearly perceived morphologically by its recent evolutionary history. 

## 4. Materials and Methods

### 4.1. Study Area and Plant Material 

Field sampling was carried out from 2012 to 2021 throughout the geographic distribution range of *Festuca indigesta* in the Iberian Peninsula. Plant material was collected in five populations (p1–p5) from three mountain ranges (The Sierra Nevada and Sierra de las Nieves National Parks, and Sierra de Gádor) located in the southeast of Spain (Figure 1). This material was then used for three purposes: (1) individuals in anthesis per population were dried and herbarium vouchers were stored in the COFC and MGC herbaria (acronyms follow Thiers [39]) for their morphometric study; (2) fresh leaves from at least five individuals were collected in plastic bags and stored at 4 °C until flow cytometric analyses, and (3) mature seeds were collected during the fruiting season (late July) for chromosome counting. Soil samples were also collected from each population for a more detailed characterization of the nature of the substrate. Data related to community ecology, type of habitat, and altitude were also obtained. 

### 4.2. Morphometric Analyses

Twelve macro-characters (culm length, culm diameter, leaf diameter, innovation auricle length, panicle length, spikelet length, number of flowers per spikelet, lower glume length, upper glume length, lemma length, awn lemma length, and anther length) and one micro-character (stomata length, i.e., accessory cells and guard cells lengths) were chosen and measured for a total of 75 specimens (i.e., 15–20 individuals per population). The characters were selected owing to their diagnostic value, especially those that, according to previous research [2,18,36] and our own observations, seemed potentially related to ploidy-variation. Three measurements were made for each character, with the exception of the stomata, for which three to seven measurements were made, and for the length of the culm and the panicle, for which only the largest size was considered. Stomatal size was measured by making epidermal impressions in the middle area of the culm on dried plants using nail polish. Floral and vegetative characters were measured under a 6.3–40× Leica stereomicroscope (model S6 D). The stomata were observed using 100–1000× light microscopy (Leica DM500). 

### 4.3. Genome Size Estimation and DNA Ploidy Level Inference

Genome size was analyzed for 31 individuals using flow cytometry. Approximately 50 mg of leaf of the sampled individual was chopped simultaneously with an equal amount of fresh leaf tissue of an internal standard using a razor blade. The internal standards chosen were *Vicia faba* “Inovec” (2C = 26.90 pg of DNA [40]) and *Bellis perennis* (2C = 3.71 pg of DNA, following calibration with *Vicia faba* Inovec, the primary reference standard). Nuclei were isolated in a Petri dish containing 1 mL of Woody Plant Buffer (WPB) [41] following the procedure described by Galbraith et al. [42]. A nuclear suspension was filtered through a 50 µm nylon filter, and 50 µg mL^−1^ of propidium iodide (PI), and 50 µg mL^−1^ of RNAse were added in order to stain the DNA and degrade the double stranded RNA, respectively. The samples were analyzed in a PartecCyFlow Space flow cytometer (Sysmex-Partec GmbH., Görlitz, Germany) equipped with a 532 nm green solid-state laser, operating at 30 mW. At least 5000 particles were analyzed per sample. Results and graphics were acquired using PartecFloMax software v2.4d (Partec GmbH, Münster, Germany). Coefficient of variation (CV) values of the G1 peak of *Festuca* of up to 5% were accepted [43], and this value was up to 4% for the standards.

The holoploid genome size (2C value, in picograms) of the *Festuca* species was estimated using the following formula: *Festuca* sp. 2C nuclear DNA content (pg) = (*Festuca* sp. G0/G1 peak mean/reference standard G0/G1 peak mean) * 2C nuclear DNA content of reference standard. The DNA ploidy level of each plant was inferred on the basis of previous chromosome counts and genome size estimates found in the Plant DNA C-values database (data.kew.org/cvalues/; accessed on 30 September 2021). 

### 4.4. Chromosome Counts

Several field-collected seeds were germinated in Petri dishes on wet filter paper at 20 °C in a growth chamber. Root tips were pre-treated with 2 mM 8-hydroxyquinoline for 24 h at 4 °C in the dark, fixed in a solution of absolute ethanol and glacial acetic acid (3:1, *v*/*v*) for at least 1 day, and stored at 4 °C until use. They were then washed with distilled water and stained with alcoholic hydrochloric acid-carmine [44] for 24–72 h at room temperature. Finally, stained meristems were squashed in 45% acetic acid. Between 25 and 50 root tips per population were analyzed in order to obtain good metaphases plates, and about 5–10 cells were counted per slide. Chromosome spreads were observed using 1000× light microscopy (Leica DM500). 

### 4.5. Soil Analyses

A total of 12 soil samples were analyzed, and at least one soil sample was collected from each population area (geographic coordinates are shown in Table 2; soils from two additional populations were also analyzed). The soil was sampled at depths of 0 to 25 cm, and the surface was previously cleaned of plant or rocks [45,46]. Soil debris were removed and sieved before analysis. The following soil analyses were carried out: determination of calcium (Ca^+2^), sodium (Na^+^), and magnesium (Mg^+2^) cations using the ICP-OES (Inductively coupled plasma-optical emission spectrometry) technique and soil acidity (pH). The presence of carbonate calcium (CaCO_3_) was detected by adding hydrochloric acid (HCl 1 M) to the soil samples [47]. All the analyses were carried in the University of Málaga’s Geomorphology and Soils IGS laboratory. The geological nature of each population area (Table 2) was consulted on the MAGNA 50 Geological Map of Spain scale 1:50,000 [48].

### 4.6. Statistical Analyses

Mean, maximum, and minimum values and the standard deviation were calculated for genome size data. A one-way ANOVA with a post hoc Tukey HSD test (*p* < 0.05) was used to test for differences in genome size. Linear regression analyses and Spearman rank correlations between holoploid genome size (2C values) and chromosome number were performed. Summary statistics (mean, maximum, and minimum values) of the morphological characters studied were also obtained in order to explore variability among cytotypes. Differences among ploidy levels and morphological character were analyzed by following the same approach as that described above. In the cases in which normality and homoscedasticity were not achieved, even after transformation, non-parametric tests were applied (Kruskal–Wallis followed by Dunn’s post-hoc tests, *p* < 0.05). A principal component analysis (PCA) was performed and plotted in order to visualize the patterns or trends of morphological separation among the different cytotypes. A total of eight variables were included, based on their usefulness regards differentiating ploidy level groups (culm length, leaf diameter, innovation auricle length, awn lemma length, number of flowers, and culm diameter were excluded). The first two eigenvectors were extracted and plotted, including the percentage of variance explained by each axis. Univariate and multivariate analyses were performed using R software (version 4.0.3).

## 5. Conclusions

The results obtained in this study provide new insights into the cytogenetic, ecological, and morphological diversity of *Festuca indigesta* subsp. *indigesta*. The taxon forms a polyploid aggregate comprising three morphologically cryptic cytotypes without any edaphic pattern in their distribution. All populations studied showed ploidy uniformity. The relationship between ploidy level and some phenotypic traits was demonstrated, but the difficulty involved in separating them using morphological data alone was also shown. Our data support the role of these mountains in southeastern Spain, where the subspecies occurs, as reservoirs of genetic diversity. This study provides a basis for further research from a phylogenetic perspective. Such integration of data will allow for elucidating the demographic history of the taxon and possible gene flow events, as well as a better understanding the role of these endemic-rich mountains in diversification processes. 

## Figures and Tables

**Figure 1 plants-11-00693-f001:**
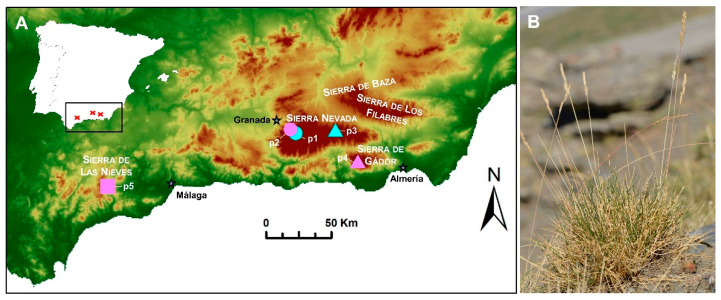
(**A**) Location of the sampled populations of *Festuca indigesta* subsp. *indigesta* in Spain (p1–p4). The three main mountain ranges (Sierra Nevada, Sierra de Gádor, and Sierra de las Nieves) where the subspecies is found are highlighted. Ploidy levels detected in each population are represented by symbols (hexaploid (circles), octoploid (triangles), and dodecaploid (square)) and substrate nature by colors (siliceous (blue) and calcareous (pink)). Variation in elevation is indicated by a gradient color map. p1, Peñones de San Francisco; p2, Collado de Las Sabinas; p3, Puerto de la Ragua; p4, Sierra de Gádor; p5, Sierra de las Nieves. Detailed information is given in Table 2. (**B**) Habit of *Festuca indigesta* subsp. *indigesta*.

**Figure 2 plants-11-00693-f002:**
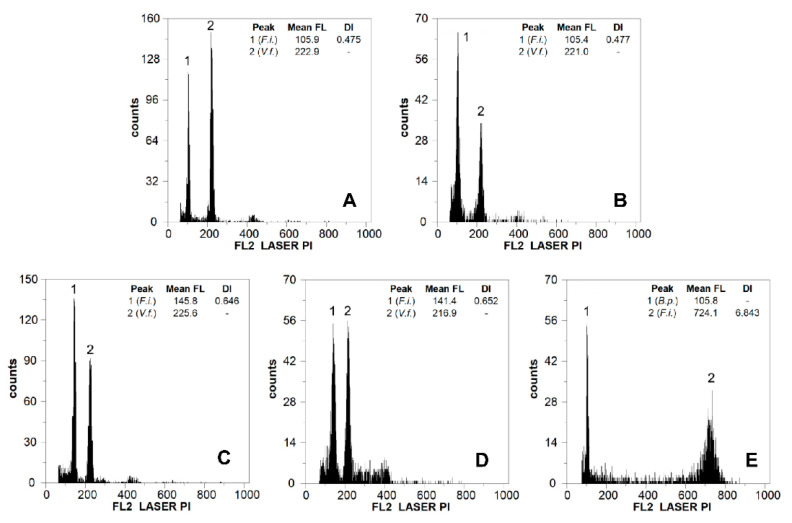
Flow cytometric histograms of relative PI fluorescence intensities for individuals of *F. indigesta* subsp. *indigesta* from (**A**) Peñones de San Francisco (p1), (**B**) Collado de Las Sabinas (p2), (**C**) Puerto de La Ragua (p3), (**D**) Sierra de Gádor (p4), (**E**) Sierra de las Nieves (p5). Mean FL, mean relative fluorescence in picograms; DI, DNA index; F.i., *F. indigesta* subsp. *indigesta*; V.f., *Vicia faba*; and B.p., *Bellis perennis*.

**Figure 3 plants-11-00693-f003:**
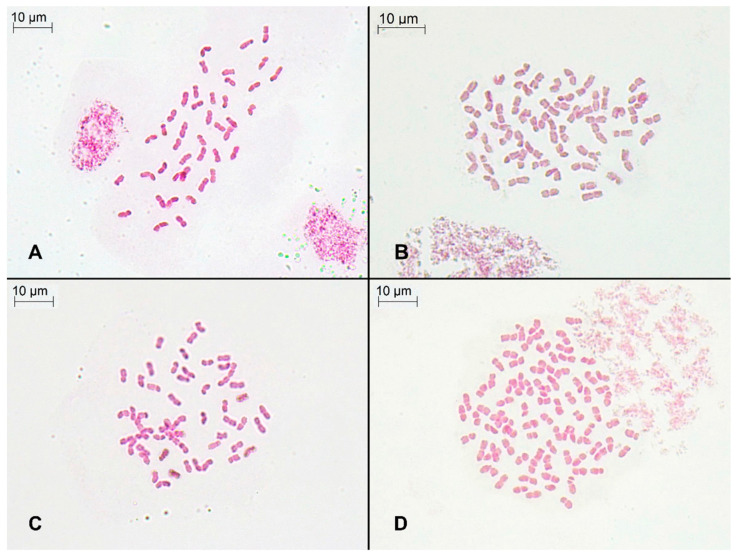
Somatic metaphase of individuals from the sampled populations: (**A**) Collado de Las Sabinas, p2 (2n = 6*x* = 42 chromosomes); (**B**) Puerto de la Ragua, p3 (2n = 8*x* = 56 chromosomes); (**C**) Sierra de Gádor, p4 (2n = 8*x* = 56 chromosomes); and (**D**) Sierra de las Nieves, p5 (2n = 12*x* = 84 chromosomes).

**Figure 4 plants-11-00693-f004:**
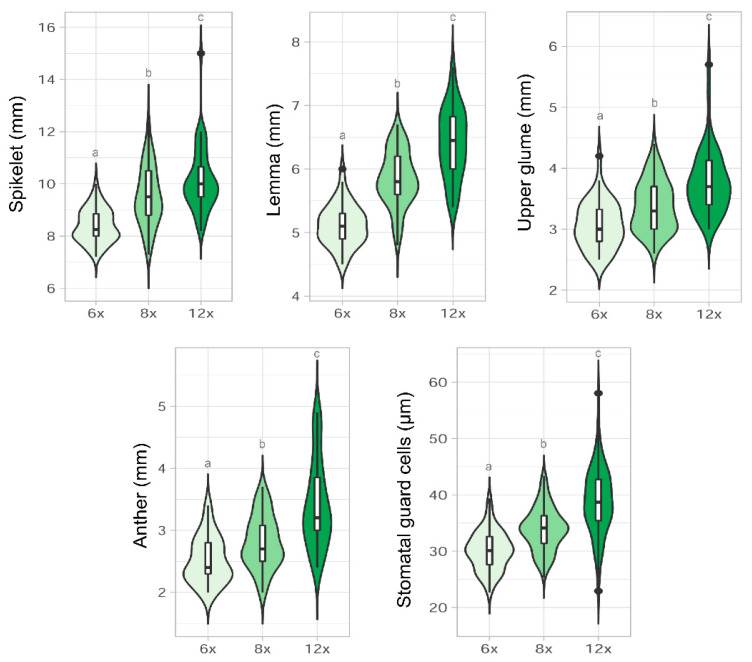
Violin plot with included boxplot of the main distinctive morphological characters, lengths of the panicle, spikelet, lemma, upper glume, anther, and stomatal guard cells, according with the different ploidy levels detected (6*x*, 8*x*, and 12*x*). Boxplots show inter-quartile range (box), median (black line within interquartile range), data range (vertical lines), and outliers (black dots). Statistically significant differences among cytotypes at *p* < 0.05 are denoted by different letters.

**Figure 5 plants-11-00693-f005:**
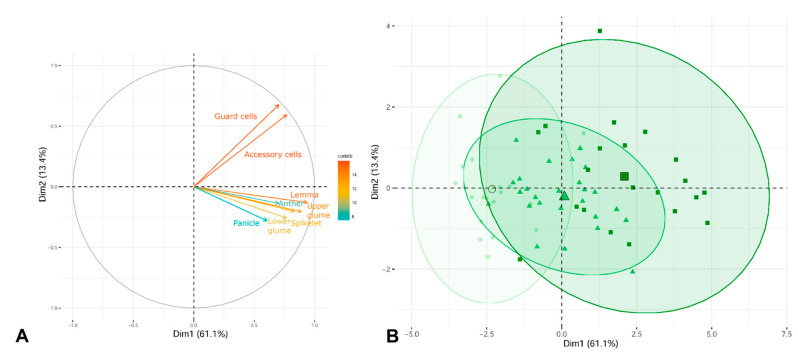
Principal components analysis (PCA). (**A**) Explanatory variables (n = 8) are shown as vectors and the most contributing variables are highlighted on the correlation plot. (**B**) Biplot of the principal component analysis PC1 vs. PC2. A 95% confidence ellipse for each grouping is shown around each group centroid (the largest symbol of each ellipse). The three different ploidy levels are represented by a green gradient and by symbols (6*x* individuals, circles; 8*x* individuals, triangles; and 12*x* individuals, squares).

**Table 1 plants-11-00693-t001:** Genome size estimates (2C values, in picograms) of the individuals analysed (n) are provided as mean and standard deviation of the mean (SD), and as minimum (Min) and maximum (Max) values. Mean coefficient of variation of the G0/G1 of sample material (CV, %) and chromosome number obtained by karyological analyses and the respective ploidy level are also provided. Different letters in genome size estimates represent statistically significant differences at *p* < 0.05.

Locality	n	2C Mean ± SD (pg)	G.s. Range (Min–Max, pg)	Cv Sample (%)	Cv Standard (%)	Chro. No.	Ploidy Level
Granada: Sierra Nevada core, Peñones de San Francisco (p1)	8	12.99 ± 0.34 ^a^	12.56–13.61	4.69	3.38	42	6*x*
Granada: Sierra Nevada core, Collado de Las Sabinas (p2)	6	13.25 ± 0.86 ^a^	12.35–14.34	4.74	3.65	42	6*x*
Granada: Sierra Nevada, Puerto de la Ragua (p3)	7	17.69 ± 0.40 ^b^	17.35–18.33	4.08	3.36	56	8*x*
Almería: Sierra de Gádor, Los Morrones (p4)	5	17.06 ± 0.44 ^b^	16.59–17.54	4.55	3.65	56	8*x*
Málaga: Sierra de las Nieves, Los Ventisqueros (p5)	5	25.33 ± 1.03 ^c^	23.73–26.37	3.83	4.00	84	12*x*

**Table 2 plants-11-00693-t002:** Analytical results of soil samples for each population. The content of Ca, Mg, and Na is referred to meq/100 g units. Coordinates, elevation, and lithologic data are provided. The lithology is based on the Geological Map of Spain (http://info.igme.es/cartografiadigital/geologica/Magna50.aspx; accessed on 23 October 2021).

Mountain Range	Population	UTM Coordinates	Elevation (m.a.s.l.)	Lithology	pH	Ca	Mg	Na	CaCO_3_ Presence
Sierra Nevada	Peñones de San Francisco (p1)	30S 465525 4105791	2543	Micaschist	6.30	3.70	0.70	2.90	-
Sierra Nevada	Collado de Las Sabinas (p2)	30S 462602 4107780	2195	Marble	8.30	7.90	1.30	2.90	-
Sierra Nevada	Puerto de La Ragua (p3)	30S 497515 4107549	2070	Micaschist	6.10	3.40	0.50	2.90	-
Sierra de Gádor	Pozo Lupión	30S 510391 4083210	1912	Limestone and dolomite	7.70	17.40	3.20	4.80	+
Sierra de Gádor	Pozo Lupión	30S 510336 4083172	1910	Limestone and dolomite	8.20	15.10	2.30	4.70	+
Sierra de Gádor	Los Morrones (p4)	30S 512984 4083216	1933	Dolomite and limestone	7.90	17.70	2.50	2.90	+
Sierra de las Nieves	Los Ventisqueros (p5)	30S 320761 4063517	1709	Limestone	7.90	27.00	0.50	4.90	+
Sierra de las Nieves	Los Ventisqueros (p5)	30S 321094 4064038	1744	Limestone	7.80	17.20	0.50	4.90	+
Sierra de las Nieves	Los Ventisqueros (p5)	30S 320843 4063724	1725	Limestone	7.40	16.30	0.70	4.80	+
Sierra de las Nieves	Los Ventisqueros (p5)	30S 322526 4064861	1595	Limestone	7.95	30.11	0.64	0.23	+
Sierra de las Nieves	Los Ventisqueros (p5)	30S 319202 4062159	1740	Limestone	8.01	21.83	4.82	0.11	+
Sierra de las Nieves	Cerro del Viento	30S 318261 4072666	1345	Dolomite	7.40	42.62	4.03	0.21	+

## Data Availability

Data is contained within the article or Appendix A.

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
