# Peer review of "Cytogenetic, Morphometric, and Ecological Characterization of Festuca indigesta Boiss. in the Southeast of Spain"

_plants, 2022, doi:10.3390/plants11050693_

Round 1

Reviewer 1 Report

The authors detected the diversity variation of Festuca indigesta  throughout its distribution range. To some extent, the morphological variation could attributed to its ploidy and genome size change. However, what caused the ploidy and genome size variation is not explained in the manuscript. For example, correlation analysis could be done to reveal the relationships between ploidy/genome size and environmental factors, such as soil conditions, altitude, latitude, longitude, precipitation, annual average temperature and so on. In addition, the accurate identification of the species Festuca indigesta is the key to further analysis. How did the authors guarantee the correct sampling? How many cells did the authors observed when does the Chromosome counts? It should be mentioned in the section '4.4. Chromosome counts'. Finally, after solving the above problems, the section '3. Discussion' should be rewrite to more in-depth indicate the relationship between species ploidy, genome size, and their corresponding morphological changes. 

Author Response

Point 1. Correlation analysis could be done to reveal the relationships between ploidy/genome size and environmental factors, such as soil conditionsaltitude, latitude, longitude, precipitation, annual average temperature and so on.

Response: The analysis of correlations between ploidy vs. environmental factors was not carried out in this revision. We consider that a greater sampling effort would be necessary for this purpose (i.e., a large-scale comparison to obtain statistically reliable correlations). We will take this suggestion into account for future studies in species with wider geographical and ecological distribution.

Point 2. How did the authors guarantee the correct sampling?

Response: Since 2011 we have been sampling the taxon in the mountains of the Iberian southeast for its taxonomic study. This has allowed us to detect a preliminary variation at morphological and ecological level. Thus, we believe that the sampling is guaranteed in terms of the purpose of the study.

Point 3. How many cells did the authors observed when does the Chromosome counts? It should be mentioned in the section '4.4. Chromosome counts'. We added the following paragraph to the section “4.4. Chromosome counts”:

Response: This data has been added in this version.

Point 4. the section '3. Discussion' should be rewrite to more in-depth indicate the relationship between species ploidy, genome size, and their corresponding morphological changes.

Response: We have tried to improve this section.

Reviewer 2 Report

Martínez-Sagarra characterized the genome size and ploidy level variation of an ecologically important grass species Festuca indigesta subsp. indigesta in Spain. The authors also tested the link between genome size and ecological features such as soil type but did not find significant correlation. Morphological variations were also detected among populations with different ploidy levels, but with overlapping trait space. The authors generated well curated flow cytometry, metaphase, and morphological trait databases, which are all valuable contributions to the scientific community. Yet given the negative results of ecology-driven patterns, molecular evidence is needed to elucidate the origin and migration of various karyotypes on these unique highland islands, which is known for richness in polyploids. This can be achieved by as simple as microsatellite markers to as elaborate as involving next generation sequencing (e.g., ddRAD seq). I found the current experiment design to be oversimplified and not well justified. For example, the rational for the correlation between soil type and cytotype is not provided in the Introduction. The authors seem to touch on this point in Line 75-85 by explaining the morphological differences between different species of the indigesta complex. But this paragraph is very confusing and I suggest the authors to rewrite it. 1) the discussion should be focused on the comparison within Festuca indigesta subsp. Indigesta rather than with other close relatives. 2) review the current knowledge on any correlation between soil type and morphotype. 3) provide more details on the morphological variation among cytotypes. Finally, I think without the knowledge from the demographic history, any conclusion will likely remain descriptive rather than explanatory. I think the authors miss an opportunity here to shed insights into the fascinating story about how these polyploids conquer these hash environments.

Minor comments:

Line 32: glacier periods of when?

Line 32: ‘one of’ instead of ‘one’

Line 33: ‘polyploidization’ instead of ‘polyploidisation’

Line 38: ‘several endemic fescues’ instead of ‘several endemisms of fescues’

Line 48: ‘two principal’ instead of ‘principal two’

Line 56: ‘consisted of’ instead of ‘formed of’

Line 81-83: the ‘pungent’ vs ‘spicy’ leaf blades are rather subtle and subjective descriptions. Can the authors provide a more quantitative way to describe the chemical features? Detailed description will also help the readers to understand the subtle differences.

Line 89: ‘regarding’ instead of ‘as regards’. Line 88-89 needs reference. If it is the data generated from the authors’ group, be clear about the source, too.

Line 100: Spell out ‘FCM’

Figure 1. What does the color mean in 1A? It looks like elevation, but it might be more appropriate to plot lithology given the focus of the study.

Figure 6. The font in A is too small to read.

Author Response

Point 1. The authors seem to touch on this point in Line 75-85 by explaining the morphological differences between different species of the indigesta complex. But this paragraph is very confusing and I suggest the authors to rewrite it.

Response: Following the reviewer's suggestion, the relationship between soil type and cytotype is provided in the Introduction section. Consequently, additional references have been included [4,5,6]. Besides, lines 75-85 have been rewritten to clarify this point.

Point 2. the discussion should be focused on the comparison within Festuca indigesta subsp. Indigesta rather than with other close relatives.

Response: We have tried to focus on the comparison of cytotypes within the species. In our opinion, it is also important to contextualize these genome size estimations with previous data for closely related species. Besides, we believe that Festuca longiauriculata needs at least a mention for being a very close diploid species with which F. indigesta subsp. indigesta co-occurs in two populations. Interestingly, in both populations the plants of F. indigesta subsp. indigesta are octoploid. Although we have not been able to study whether there has been any hybridization process between both taxa, we find it interesting to present this information for future studies.

Point 3. review the current knowledge on any correlation between soil type and morphotype.

Response: No morphological pattern was observed with respect to the type of substrate (e.g., the smallest lemma and stomata sizes were found both in plants that grow in siliceous substrate, such as Peñones de San Francisco population, and in calcareous substrate, such as Las Sabinas population).

3) provide more details on the morphological variation among cytotypes. Response: Following this recommendation, we have tried to improve this point.

Finally, I think without the knowledge from the demographic history, any conclusion will likely remain descriptive rather than explanatory. I think the authors miss an opportunity here to shed insights into the fascinating story about how these polyploids conquer these hash environments. This is a very interesting question that ourselves pose when discussing our results.

Response: We found the reviewer's suggestion/recommendation very interesting. We believe that the present study can be considered a stepping stone for future molecular analyses (see Conclusion section).

Minor comments:

Line 32: glacier periods of when? Done (Pleistocene glacial periods)

Line 32: ‘one of’ instead of ‘one’ Done

Line 33: ‘polyploidization’ instead of ‘polyploidisation’ Done

Line 38: ‘several endemic fescues’ instead of ‘several endemisms of fescues’ Done

Line 48: ‘two principal’ instead of ‘principal two’ Done

Line 56: ‘consisted of’ instead of ‘formed of’ Done

Line 81-83: the ‘pungent’ vs ‘spicy’ leaf blades are rather subtle and subjective descriptions. Can the authors provide a more quantitative way to describe the chemical features? Detailed description will also help the readers to understand the subtle differences. We clarified this point in the text.

Line 89: ‘regarding’ instead of ‘as regards’. Done.

Line 88-89 needs reference. If it is the data generated from the authors’ group, be clear about the source, too. We referred to our analyses, this has been clarified in the text.

Line 100: Spell out ‘FCM’ Done

Figure 1. What does the color mean in 1A? It looks like elevation, but it might be more appropriate to plot lithology given the focus of the study. We clarified this in the figure legend.

Figure 6. The font in A is too small to read. Following this suggestion, we increased the font size in Figure 6.

Reviewer 3 Report

The present paper describes cytogenetic, morphological and ecological characteristics of Festuca indigesta in relation to environmental conditions in the southeast region in Spain. The paper provides significant information on the characteristics of its distribution. It seems valuable to be published in the present journal. However, there are some uncertain, not understandable descriptions in the text. Revision would be needed. The reviewer gives some comments and questions as follows:

Comments:

1) The title is grammatically strange. The word ‘integrating’ needs some words after ‘into’ or ‘with’, such as “Integrating … into (or with) ….” Thus, changing the title is recommended, for example, “Cytogenetic, morphometric, and ecological characterization of Festuca indigesta in the southeast of Spain.” It might be simple and clear.

2) Line 40: In “the Festuca species and those of many other genera”, what does ‘those’  indicate? It might be “… and the species of many other genera”.

3) Line 108: The data for the standard are not shown in Table 1.

4) Figure 3 is a duplication of Table 1. It is not necessary.

5) Figure 4. Quality of chromosome pictures are low. The chromosomes are too small, and spaces around the chromosomes are large. Enlarge the part of the chromosomes. Show the chromosomes with black and white with higher contrast.

6) Line 184: In the scientific paper, we do not use the words “in general”. In the present case, using “As a whole” would be better.

7) Line 188: What is the meaning of “differential characters”?

8) Figure 5: Explanation of the figure is not enough. Add the explanation on a, b, and c, and boxes and bars in the figures. What are the black dots?

9) Figure 6: Legend of the figure is not the title of the Figure 6. Title of the figures should be shown. It is written in the line 215 that “the hexaploids and dodecaploids were clearly separated, forming mostly non-overlapping groups”, however, in the Figure 6B, the ellipses for hexaploids and dodecaploids are overlapped. The explanation should be changed. Explanation for a large circle, triangle and square should be shown.

10) Line 228: “a well-represented octoploid level was detected” is not understandable.

11) Line 262-265: The sentence structure is not understandable.

12) Line 264: What is the meaning of ‘taxon-dominant’?

13) Line 265: What does the word ‘they’ indicate?

14) Line 362: What is alcoholic acetic-carmine? Carmine is not solved in alcohol, but acetic acid solution. What is the percentage of carmine and which product?

Correction recommended:

Line 13: … until recently it was considered that its range of distribution was restricted to …. However, it has been recently extended to other calcareous mountains.

Line 16: Genome sizes and ploidy levels were …

Line 18: … substrate, i.e., pH, and … contents.

Line 20: Hidden The cytogenetic …

Line 36: the Iberian Peninsula.

Line 38: … and the others that are …

Line 40: … and those the species of

Line 68: … where the subspecies was found are …

Line 77: … margins, and …

Line 78: … cross-section. among other traits.

Line 98: Considering the facts mentioned above, we decided to assess genome size …

Line 102: and to clarify whether there was …

Line 120: … among the populations …. Thus, Plants from …

Line 124: … (Table 1). Figure 3).

Line 135: It was also confirmed by chromosome counts that these … to the three ploidy levels (Figure 4): …

Line 141: … in chromosome numbers was found, … size estimates. that could … level.

Line 146: p2(2n=6x=42); … p3(2n=8x=56); … p4(2n=8x=56) and D) …, p5(2n=12x=84). n should also be italic.

Line 150: … contents …

Line 151: … could be grouped …

Line 152: … were acidic …

Line 154: …, as could be …

Line 155: … were alkaline …

Line 158: …, as was typical …

Line 159: … the soils were also … but had a …

Line 161: … Ca and Mg contents in the …

Line 166: … was a part of …

Line 173: … vegetations were found where, as well as F. indigesta subsp. indigesta, Vella spinose Boiss, … were growing.

Line 175: The subspecies (?): Which subspecies?

Line 184: In general, As a whole, … characters increased …

Line 185: …, although considerable overlaps … were detected ….

Line 186: … among the cytotypes … studied (Figure 5), …

Line 203: … are shown as …

Line 223: Our study provides additional data to those … supported with flow cytometry as a reliable method. with … genus.

Line 231: … the octoploid plants were found to a greater extent than previously assumed, as we found octoploid plants in all the samples from … no data have been available ….

Line 237: Fuente et al. [17] speculated …

Line 241: Not understandable “Be that as it may”.

Line 244: The dodecaploid plants are new to the species and were found …

Line 247: The decaploid plants that would be …

Line 255: … octploids were …

Line 256: …, which had a broader altitudinal adaptability.

Line 259: … any patterns …

Line 263: factors are …

Line 264: which are …   

Line 272: … differences among cytotypes …

Line 280: have

Line 293: range

Line 367-377: The word ‘population’ is used incorrectly. It must be ‘population area’ or ‘population areas’.

Author Response

1) The title is grammatically strange. The word ‘integrating’ needs some words after ‘into’ or ‘with’, such as “Integrating … into (or with) ….” Thus, changing the title is recommended, for example, “Cytogenetic, morphometric, and ecological characterization of Festuca indigesta in the southeast of Spain.” It might be simple and clear.

Response: We closely followed this recommendation and the title has been changed.

2) Line 40: In “the Festuca species and those of many other genera”, what does ‘those’  indicate? It might be “… and the species of many other genera”.

Response: Done.

3) Line 108: The data for the standard are not shown in Table 1.

Response: In this version, data for standard were added in Table 1 (CV standard).

4) Figure 3 is a duplication of Table 1. It is not necessary.  We removed the Figure 3.

5) Figure 4. Quality of chromosome pictures are low. The chromosomes are too small, and spaces around the chromosomes are large. Enlarge the part of the chromosomes. Show the chromosomes with black and white with higher contrast.

Response: The size of the figure has been increased in this revision. However, we have kept the color in the figure because black and white with higer contrast does not look nice for chromosome counting.

6) Line 184: In the scientific paper, we do not use the words “in general”. In the present case, using “As a whole” would be better.

Response: Done

7) Line 188: What is the meaning of “differential characters”?

Response: It was clarified in the text.

8) Figure 5: Explanation of the figure is not enough. Add the explanation on a, b, and c, and boxes and bars in the figures. What are the black dots?

Response: Done

9) Figure 6: Legend of the figure is not the title of the Figure 6. Title of the figures should be shown. It is written in the line 215 that “the hexaploids and dodecaploids were clearly separated, forming mostly non-overlapping groups”, however, in the Figure 6B, the ellipses for hexaploids and dodecaploids are overlapped. The explanation should be changed. Explanation for a large circle, triangle and square should be shown.

Response: We agree with the reviewer and we re-phrased this sentence. The largest symbol of each ellipse indicates the centroid (see legend Figure 5)

10) Line 228: “a well-represented octoploid level was detected” is not understandable.

Response: We re-phrased this sentence.

11) Line 262-265: The sentence structure is not understandable.

Response: In order to clarify this point, we re-phrased this sentence.

12) Line 264: What is the meaning of ‘taxon-dominant’?

Response: We re-phrased this sentence.

13) Line 265: What does the word ‘they’ indicate?

Response: We have rewritten this sentence.

14) Line 362: What is alcoholic acetic-carmine? Carmine is not solved in alcohol, but acetic acid solution. What is the percentage of carmine and which product? Response: We clarified the correct chemical solution, “alcoholic hydrochloric acid-carmine”. We follow the protocol of Snow (1963), in which the stain is made by boiling 4 mg of carmine in 15 ml of distilled water. After cooling, 95 ml of 85% alcohol is added, and the solution filtered.

Correction recommended:

Line 13: … until recently it was considered that its range of distribution was restricted to …. However, it has been recently extended to other calcareous mountains. Done

Line 16: Genome sizes and ploidy levels were … Done

Line 18: … substrate, i.e., pH, and … contents. Done

Line 20: Hidden The cytogenetic … Done

Line 40: … and those the species of Done

Line 68: … where the subspecies was found are … Done

Line 77: … margins, and … Done

Line 78: … cross-section. among other traits.  We re-phrased this sentence

Line 102: and to clarify whether there was … Done

Line 120: … among the populations …. Thus, Plants from … Done

Line 135: It was also confirmed by chromosome counts that these … to the three ploidy levels (Figure 4): … Done

Line 141: … in chromosome numbers was found, … size estimates. that could … level. Done

Line 150: … contents … Done

Line 151: … could be grouped … Done

Line 152: … were acidic … Done

Line 155: … were alkaline … Done

Line 159: … the soils were also … but had a … Done

Line 161: … Ca and Mg contents in the … Done

Line 175: The subspecies (?): Which subspecies? Done (Festuca indigesta subsp. indigesta).

Line 184: In general, As a whole, … characters increased … Done

Line 185: …, although considerable overlaps … were detected …. We re-phrased this sentence

Line 186: … among the cytotypes … studied (Figure 5), … Done

Line 203: … are shown as … Done

Line 223: Our study provides additional data to those … supported with flow cytometry as a reliable method. with … genus. Done

Line 231: … the octoploid plants were found to a greater extent than previously assumed, as we found octoploid plants in all the samples from … no data have been available …. Done

Line 237: Fuente et al. [17] speculated … Done

Line 244: The dodecaploid plants are new to the species and were found … We prefer to refer to dodecaploid level in this sentence.

Line 263: factors are … Done

Line 264: which are … we re-phrased this sentence.

Line 272: … differences among cytotypes … Done

Line 280: have Done

Line 293: range Done

Line 367-377: The word ‘population’ is used incorrectly. It must be ‘population area’ or ‘population areas’. Done

Line 124: … (Table 1). Figure 3). Done

Line 256: …, which had a broader altitudinal adaptability. Done

Line 259: … any patterns … Done

Line 241: Not understandable “Be that as it may”. We re-phrased this sentence.

Line 173: … vegetations were found where, as well as F. indigesta subsp. indigestaVella spinose Boiss, … were growing. We modified this sentence (the accompanying plants text has been removed) following the recommendation of another reviewer.

The following modifications imply a change of meaning of the sentence, so they were not applied:

Line 36: the Iberian Peninsula.

Line 38: … and the others that are …

Line 98: Considering the facts mentioned above, we decided to assess genome size …

Line 146: p2(2n=6x=42); … p3(2n=8x=56); … p4(2n=8x=56) and D) …, p5(2n=12x=84). n should also be italic.

Line 158: …, as was typical …

Line 166: … was a part of …

Line 247: The decaploid plants that would be …

Line 255: … octploids were …

Reviewer 4 Report

Plants 1586508

The above paper examines the cytology, morphology and ecology of different groups of Festuca indigesta (Poaceae) in southeast Spain.

The experimental work and analysis appear sound, and the paper is generally well written. The manuscript is suitable for publication in the Journal.

Some suggestions for the authors are indicated below.

Figure 2 is redundant and could be deleted.  The important information on sample variance is provided in Table 1.

Figure 3 should be deleted since the data on 2C values are provided in Table 1.

Table 2.  Indicate the number of soil samples collected for each site (N = ?).

Table 2.  Delete the column on the presence of CaCO3 (information on Lithology is sufficient).

Pages 5 and 6.  Delete the information on companion species which doesn’t appear to be relevant to the study on Festuca.

Legend to Figure 5. Indicate what the letters (a, b, and c) refer to.

Figure 6 and the first paragraph on page 8 (PCA) should be deleted.  The data shown in Figure 5 already provide an overview of the variation in plant morphology.

The ‘Conclusion’ is too long.  Reduce this section to 3-4 sentences, highlighting the main findings of the study.

Author Response

Point 1. Figure 2 is redundant and could be deleted.  The important information on sample variance is provided in Table 1.

Response: We have not followed the reviewer’s recommendation at this point as we consider Figure 2 to be important.

Point 2. Figure 3 should be deleted since the data on 2C values are provided in Table 1.

Response: Figure 3 has been deleted following reviewer recommendation

Point 3. Table 2.  Indicate the number of soil samples collected for each site (N = ?).

Response: The total number of soil samples is indicated in Material and Methods section (each row of the table corresponds to a soil sample).

Point 4. Table 2.  Delete the column on the presence of CaCO(information on Lithology is sufficient).

Response: Due to major changes these suggestions are no longer applicable

Point 5. Pages 5 and 6.  Delete the information on companion species which doesn’t appear to be relevant to the study on Festuca.

Response: We agree with the reviewer and the accompanying species information has been removed. However, we have considered leaving some general remarks on the ecology of the species. If the reviewer does not think it is apporpiate, we can remove it.

Point 6. Legend to Figure 5. Indicate what the letters (a, b, and c) refer to.

Response: Done

Point 7. Figure 6 and the first paragraph on page 8 (PCA) should be deleted.  The data shown in Figure 5 already provide an overview of the variation in plant morphology.

Response: We believe that the figure 6 shows valuable information to visualize the morphological separation tendencies (including 8 of the most important taxonomic characters of the genus) among cytotypes, so we have decided to keep it.

Point 8. The ‘Conclusion’ is too long.  Reduce this section to 3-4 sentences, highlighting the main findings of the study.

Response: We have shortened the Conclusion as much as possible, but we have included another reviewer's comments.

Round 2

Reviewer 3 Report

The manuscript is revised properly as the reviewer pointed out, and the explanations are effectively added to the original one. The reviewer considers that the paper would be valuable to be accepted for publication.